# Comparative Analysis of the Effect of Inorganic and Organic Chemicals with Silver Nanoparticles on Soybean under Flooding Stress

**DOI:** 10.3390/ijms21041300

**Published:** 2020-02-14

**Authors:** Takuya Hashimoto, Ghazala Mustafa, Takumi Nishiuchi, Setsuko Komatsu

**Affiliations:** 1Faculty of Environment and Information Sciences, Fukui University of Technology, Fukui 910-8505, Japan; regios2400@yahoo.co.jp (T.H.); mghazala@qau.edu.pk (G.M.); 2Department of Plant Sciences, Quaid-i-Azam University, Islamabad 45320, Pakistan; 3Institute for Gene Research, Kanazawa University, Kanazawa 920-8640, Japan; tnish9@staff.kanazawa-u.ac.jp

**Keywords:** proteomics, soybean, flooding, silver nanoparticles, chemicals

## Abstract

Extensive utilization of silver nanoparticles (NPs) in agricultural products results in their interaction with other chemicals in the environment. To study the combined effects of silver NPs with nicotinic acid and potassium nitrate (KNO_3_), a gel-free/label-free proteomic technique was used. Root length/weight and hypocotyl length/weight of soybean were enhanced by silver NPs mixed with nicotinic acid and KNO_3_. Out of a total 6340 identified proteins, 351 proteins were significantly changed, out of which 247 and 104 proteins increased and decreased, respectively. Differentially changed proteins were predominantly associated with protein degradation and synthesis according to the functional categorization. Protein-degradation-related proteins mainly consisted of the proteasome degradation pathway. The cell death was significantly higher in the root tips of soybean under the combined treatment compared to flooding stress. Accumulation of calnexin/calreticulin and glycoproteins was significantly increased under flooding with silver NPs, nicotinic acid, and KNO_3_. Growth of soybean seedlings with silver NPs, nicotinic acid, and KNO_3_ was improved under flooding stress. These results suggest that the combined mixture of silver NPs, nicotinic acid, and KNO_3_ causes positive effects on soybean seedling by regulating the protein quality control for the mis-folded proteins in the endoplasmic reticulum. Therefore, it might improve the growth of soybean under flooding stress.

## 1. Introduction

Industrial revolution drastically increased atmospheric concentration of carbon dioxide that leads to global warming and changed the precipitation pattern [1]. Changing climatic conditions caused tragic losses in crop productivity [2]. Under these changing climatic conditions, plants are at the forefront of different kinds of abiotic stresses including drought [3], cold [4,5], salinity [6], and heat stress [2]. Flooding is one of the most widely spread abiotic stresses that affects all the terrestrial plants by limiting the carbon dioxide, oxygen, ethylene, and nitric oxide those play important roles in signal transduction cascades [7]. Flooding also affects the soil chemical characteristics including soil pH and redox potential [8]; due to which the availability of soil nutrients is hindered, which results in accumulation of phytotoxins [9]. Flooding itself acts as a complex stress that hampers the plant growth [10]. Based on these reports, flooding severely reduced crop growth and productivity.

Soybean is one of the most important legume crops due to its high protein level and oil contents. Soybean is important for biodiesel production because it emits zero nitrogen, which is highly beneficial [11]. Its yield was decreased by 17–43% at vegetative stage and 50–56% at reproductive stage due to flooding [12]. Yield reduction was due to various changes caused by flooding at seedling stage in soybeans [13]. At the early stage, flooding induced the alcohol fermentation and ethylene biosynthesis-related proteins in soybean [14]. At the seedling stage of soybean, flooding stress caused sucrose accumulation, cell wall loosening, mitochondrial impairment, and proteasome-mediated proteolysis [15]. Soybean is highly susceptible to flooding stress especially at the early stage.

One of the nanoparticles (NPs), silver NPs, enhanced shoot and root length, leaf area, and biochemical attributes such as chlorophyll, carbohydrate, protein contents, and antioxidant enzymes in *Brassica juncea* [16,17]. Silver NPs enhanced seed germination and seedling growth in *Boswellia ovalifoliolata* and reduced the root length in *Arabidopsis thaliana* [18,19]. In *Eruca sativa*, silver NPs stimulated plant growth [20]. Transcription of antioxidant and aquaporin genes was altered by silver NPs in *A. thaliana* [21]. In soybean, silver NPs helped to mitigate the flooding stress condition by regulating the proteins related to fermentation, glycolysis, amino acid synthesis, and wax formation [22,23]. Silver NPs caused variable effects on plants; therefore, the molecular mechanisms underlying these effects demand investigation.

Within the environment, NPs come in contact with other materials like chemical compounds [24,25]. These interactions could be of additive, synergistic, and antagonistic type [26]. Nicotinic acid is involved in the primary and secondary metabolism of plants. It acts as a building block for pyridine compounds like trigonelline, nicotine, anabasin, and ricinine [27]. Nicotinamide disturbs the cytosolic adenosine triphosphate concentration, and regulated period length adjustment, meristem activation, and root growth [28]. Application of nicotinic acid protected spruce seedlings from weevil attack through epigenetic regulation [29]. Nicotinic acid reduced the oxidative stress in plant cells by regulating the aconitase, fumarase, and glutathione metabolism [30]. On the other hand, potassium nitrate (KNO_3_) is an inorganic chemical used for seed priming and dormancy breaking in tomato and maize [31,32]. Potassium nitrate reduced the germination time and enhanced the germination rate in tomato by regulating the nitrate reductase, catalase, and superoxide dismutase activities [31]. Individual application of organic/inorganic chemicals and NPs accelerated the growth of different plants; however, molecular mechanisms altered by their combined applications are still not clear. In the present study, the alterations induced by the silver NPs with nicotinic acid and KNO_3_ were evaluated using gel-free/label-free proteomic technique. In addition, molecular and biochemical analyses were performed to confirm the proteomics results.

## 2. Results

### 2.1. Growth Response of Soybean to Silver NPs Mixed with Organic and Inorganic Chemicals

In order to investigate the effects of silver NPs with and without nicotinic acid and KNO_3_, two-day-old soybeans were treated without or with 5 ppm silver NPs, 8 µM nicotinic acid, 0.1 mM KNO_3_, and 5 ppm silver NPs/8 µM nicotinic acid/0.1mM KNO_3_. After treatments, root length/weight and hypocotyl length/weight were measured at two days of stress. Root length and weight were increased under the silver NPs/nicotinic acid/KNO_3_ treatment compared to control (Figure 1). On the other hand, the hypocotyl length was significantly increased under the silver NPs/nicotinic acid/KNO_3_ treatment compared to other treatments and control. Hypocotyl length was increased with silver NPs, nicotinic acid, or KNO_3_; however, change in hypocotyl length was insignificant among silver NPs, nicotinic acid, and KNO_3_ treatments (Figure 1). Hypocotyl weight was significantly increased under the silver NPs/nicotinic acid/KNO_3_ treatment compared to control and all other treatments; however, change in hypocotyl weight was insignificant among silver NPs, KNO_3_ treatments, and nicotinic acid (Figure 1).

### 2.2. Protein Responses in Soybean to Silver NPs Mixed with Organic and Inorganic Chemicals

To get a deep insight into the effects caused by silver NPs mixed with nicotinic acid and KNO_3_ on soybean root under flooding stress, a gel-free/label-free proteomic technique was used. In total, 6340 proteins were identified in the proteomics analysis. In the differential analysis, 351 proteins were significantly changed. Out of these 351 proteins, 247 and 104 proteins increased and decreased, respectively (Appendix A, Figure 2).

To determine the functional role of these proteins, functional categorization was performed using MapMan bin codes. The majority of the significantly changed proteins in the soybean root, treated with silver NPs mixed with nicotinic acid and KNO_3_ compared to control, were related to protein (60 proteins), stress (25 proteins), transport (14 proteins), RNA (19 proteins), and cell wall (12 proteins) (Figure 2). The differentially changed proteins related to the protein category were further divided into sub-categories. This sub-categorization revealed that more proteins related to proteins degradation were significantly changed; out of which, the abundance of 14 and 6 proteins was increased and decreased, respectively. Among the identified protein degradation-related proteins, RRM domain-containing protein was accumulated at a higher degree than the proteasome subunit alpha and ubiquitin conjugating two domain-containing proteins (Appendix A). The second most changed proteins related to protein category were related to protein synthesis. In total, 18 protein synthesis-related proteins were identified; out of which, the abundance of 12 and 6 proteins was decreased and increased, respectively (Figure 2).

The second most significantly changed protein category was stress. From the stress-related proteins, a total of 25 proteins were significantly changed, out of which 17 proteins increased in abundance while 8 proteins decreased. Among the identified stress-related proteins, the abundance of methyltransferases increased; however, the abundance of MLO like protein decreased. The third most significantly changed protein category was transport. From the transport category, a total of 14 proteins were significantly changed, out of which 13 proteins increased in abundance while one protein decreased. Among the identified transport-related proteins, aldo-ket-red-domain-containing protein increased; however, uncharacterized protein decreased (Appendix A). Principal component analysis (PCA) data of total proteins from six samples under control and mixture treatment depicted the closeness of the three independent biological replicates (Appendix A).

### 2.3. Evaluation of Root-Tip Cell Death in Flooded Soybean Seedlings

To investigate the role of protein degradation-related proteins in the flooding stressed soybean seedlings treated with or without silver NPs/nicotinic acid/KNO_3_, cell death analysis was performed. From the functional categorization of the significantly changed proteins, 60 proteins related to the protein category were identified, 20 of which were involved in protein degradation as part of the ubiquitin proteasome degradation pathway (Figure 2). Two-day-old soybeans were flooded with or without silver NPs/ nicotinic acid/ KNO_3_ and stained with Evans-blue dye to assess the cell death (Figure 3). The degree of staining was higher in the root tip of flooded soybean compared to control; however, it was much higher in root tip treated with silver NPs/ nicotinic acid/ KNO_3_ under flooding stress compared to silver NPs/ nicotinic acid/ KNO_3_, flooded, and untreated soybean (Figure 3). There was no difference in the stain uptake between flooded and silver NPs/ nicotinic acid/ KNO_3-_treated soybean.

### 2.4. Calreticulin/Calnexin Cycle and Glycolysis in Flooded Soybean Seedlings

To analyze the changes in the accumulation pattern of calreticulin and calnexin under flooding stress with or without silver NPs/ nicotinic acid/ KNO_3_, immuno-blot analysis was performed. Two-day-old soybeans were flooded with or without silver NPs/nicotinic acid/ KNO_3_, and proteins were extracted from root at two days of stress. Extracted proteins were separated on SDS-PAGE, transferred onto membranes, and cross reacted with calnexin, calreticulin, and ConcanavalinA antibodies (Figure 4 and Figure 5). The relative band intensities were calculated.

Immuno-blot results identified two bands for calnexin. One was of 63 kDa and other was 45 kDa. In the first identified band, the accumulation of the calnexin was not significantly changed under untreated and treated samples. On the other hand, the accumulation of the second band significantly increased in soybean root under flooding stress with or without silver NPs/ nicotinic acid/KNO_3_ treatment compared to untreated and silver NPs/nicotinic acid/KNO_3_ treatment (Figure 4). In the immuno-blot analysis, the band for calreticulin appeared at 55 kDa. The accumulation of calreticulin significantly increased under flooding with silver NPs/nicotinic acid/KNO_3_ treatment compared to untreated and silver NPs/nicotinic acid/KNO_3_ treatment. It remained unchanged under flooding and silver NPs/ nicotinic acid/ KNO_3_ treatment compared to control (Figure 4).

From the functional categorization of the identified proteins, protein synthesis-related proteins were significantly changed under silver NPs/ nicotinic acid/ KNO_3_ treatment (Appendix A, Figure 2). Based on these results, in order to check the protein synthesis, immuno-blot analysis was performed to investigate the level of glycosylation. The accumulation of glycoproteins was increased under flooding with silver NPs/nicotinic acid/KNO_3_ treatment compared to control; however, it remained unchanged under flooding, silver NPs/nicotinic acid/ KNO_3_, and control (Figure 5).

### 2.5. Growth Response of Soybean to Silver NPs Mixed with Organic and Inorganic Chemicals under Flooding Stress

From the immuno-blot results, the accumulation of glycoproteins was increased under flooding with silver NPs/nicotinic acid/ KNO_3_ treatment compared to control (Figure 5); therefore, the growth of soybean seedlings under flooding stress with or without silver NPs/nicotinic acid/KNO_3_ treatment was analyzed. Two-day-old soybeans were flooded with or without silver NPs/ nicotinic acid/KNO_3_ for two days. After treatments, length and weight of root including hypocotyl were measured (Figure 6). The length and weight of root including hypocotyl significantly increased under flooding with silver NPs/nicotinic acid/ KNO_3_ treatment compared to flooding stress alone (Figure 6).

## 3. Discussion

### 3.1. Soybean Morphology Alters under Silver NPs, Nicotinic Acid, and KNO_3_

To investigate the effects caused by silver NPs mixed with nicotinic acid and KNO_3_ on soybean growth (Figure 1), soybean was treated with or without silver NPs mixed with nicotinic acid and KNO_3_. Nanoparticles combined with different compounds caused variable effects on plants, which could be antagonistic, synergistic, or additive. Promotion of seed germination and inhibition of root growth of *Lepidium sativum*, flax, cucumber, and wheat were more under individual NPs compared to the combined mixture of copper and zinc NPs [33]. Mixture of copper and zinc NPs inhibited the growth of common bean compared to the individual application of NPs [34]. Binary mixture of copper and zinc NPs significantly reduced plant fresh weight and root length in spinach compared to individual NPs due to increased internal uptake of copper and zinc inside the cells [35]. Binary mixture of copper and zinc oxide NPs caused deleterious effects on radish seeds and inhibited germination. Moreover, the effect of binary mixture of copper and zinc NPs in radish root/shoot length was less toxic than individual copper NPs and more toxic than individual zinc NPs [36]. Binary mixtures of NPs cause opposite or combined effects to the individual NPs.

Zinc NPs combined with tetrabromobisphenol caused severe toxicity to the freshwater microalgae by causing accumulation of reactive oxygen species that leads to oxidative stress [37]. Gold NPs alone caused less toxicity in microalgae; however, when combined with microplastics, the toxicity increased [38]. Silver NPs combined with nicotinic acid and KNO_3_ improved the growth and development of wheat seedlings [39]. In the present study, growth of soybean seedling significantly increased under the combined application of silver NPs, nicotinic acid, and KNO_3_ (Figure 1). Nicotinic acid increased the plant growth by increasing the photosynthesis [40]; while KNO_3_ is important for its role in photosynthesis, enzyme activation, and stress resistance [41]. In wheat, the silver NPs combined with nicotinic acid and KNO_3_ maintained the redox homeostasis through regulation of glycolysis and increased activities of antioxidant enzymes that regulated energy metabolism [39]. These results suggest that silver NPs synergistically enhance soybean-plant growth with nicotinic acid and KNO_3_ similar to the wheat. Although growth of soybean seedling significantly increased under the combined application of silver NPs, nicotinic acid, and KNO_3_, the detailed mechanism of soybean under the combined treatment of silver NPs, nicotinic acid, and KNO_3_ is still not clear. In this research, proteomic technique as well as molecular and biochemical analyses were performed to confirm the mechanism.

### 3.2. Role of Proteasome Degradation Proteins under Silver NPs, Nicotinic Acid, and KNO_3_

To get an understanding of the proteins response under the silver NPs mixed with nicotinic acid and KNO_3_ on soybean growth under flooding stress, the identified proteins were subjected to functional categorization (Figure 2). Within cells, protein synthesis and degradation are well balanced, because a small decrease in synthesis or increase in degradation results in cell death [42]. Similarly, in the present study, the abundance of protein degradation related proteins increased (Figure 2). Under stress conditions, cells maintain the cellular integrity by repairing the stress-induced damaged proteins by degradation. This degradation is important to restrict the accumulation of misfolded or heavily damaged proteins [43]. The ubiquitin proteasome system acts as a regulatory mechanism for the protein regulation in plants. Various studies support the evidence that ubiquitin proteasome system is primarily regulated under the stressful conditions [44]. The ubiquitin proteasome pathway facilitates the plants to alter their proteins to effectively combat the stress conditions. This regulation is an integral part of plant adaptation to the environmental stress. The ubiquitin proteasome system responds to the changing environmental conditions including stress by regulating the protein degradation [45]. In soybean, flooding stress induces ubiquitin/proteasome-mediated proteolysis, which leads to the degradation and loss of the root tip [46]. The ubiquitin proteasome system responds to the stress conditions and mediates the protein regulation.

The ubiquitin proteasome system helps the plant to increase the oxidative stress tolerance by altering its proteins so that it efficiently perceives and responds to the stress [47]. In *Arabidopsis*, 26S proteasome regulatory particle mutants increased oxidative stress tolerance. In the present study, several proteins related to protein degradation, particularly ubiquitin-proteasome-related proteins increased under silver NPs mixed with nicotinic acid and KNO_3_ treatment (Figure 2, Appendix A). Moreover, cell death increased in the soybean root-tip under silver NPs mixed with nicotinic acid and KNO_3_ (Figure 3). In soybean, ascorbate peroxidase was suppressed under flooding stress [48]; thereby, it induced the oxidative stress [49]. These results suggest that ubiquitin proteasome system is regulated under silver NPs mixed with nicotinic acid and KNO_3_ by increasing the oxidative stress tolerance in order to maintain the cellular integrity; suggesting that growth might be improved under flooding stress.

### 3.3. Calnexin/Calreticulin Cycle and Glycolysis Have Important Role in Soybean Seedling under Flooding with Silver NPs, Nicotinic Acid, and KNO_3_

In order to investigate the level of protein synthesis in soybean root under the silver NPs mixed with nicotinic acid and KNO_3_, the accumulation of calnexin/calreticulin and glycoproteins were accessed (Figure 4 and Figure 5). Endoplasmic reticulum mediates the protein folding and assembly through a well-coordinated system of chaperones including calnexin, protein disulfide isomerase, and heat shock proteins [50]. Calnexin is involved in the protein folding and quality control [51]. In soybean, calnexin was increased during the first day while it decreased at the two-day flooding stress [15,52]. In soybean, calnexin and calreticulin decreased and it led to the reduced accumulation of glycoproteins and disruption of endoplasmic reticulum homeostasis under flooding and drought stresses [53]. On the other hand, heterologous expression of rice calnexin confers drought tolerance in tobacco [54]. Calreticulin, a major calcium binding chaperone in the endoplasmic reticulum, is a key component of the calreticulin/calnexin cycle, which is responsible for the folding of newly synthesized proteins, especially glycoproteins, for quality control and stability in the endoplasmic reticulum [55]. In the present study, accumulation of calnexin and calreticulin was higher in the soybean under the silver NPs mixed with nicotinic acid and KNO_3_ compared to the control (Figure 4). These results suggest that calnexin/calreticulin cycle is enhanced with the silver NPs mixed with nicotinic acid and KNO_3_ in order to regulate the misfolded proteins under flooding stress.

Protein glycosylation is an important post-translational modification with roles in biological activities of proteins, induction of correct folding, signal recognition, protein structure stability, protein interactions, and ultimately the growth of plant organs [56,57,58]. In order to maintain the protein quality control, endoplasmic reticulum is responsible for the protein degradation of misfolded proteins under stressful conditions [59]. The activity of ubiquitin conjugating enzyme enhances the endoplasmic reticulum for the degradation of misfolded proteins [60]. Similarly, in the present study, the ubiquitin conjugating protein increased under silver NPs mixed with nicotinic acid and KNO_3_ (Figure 2 and Appendix A). The N-glycan degradation of glycoproteins in endoplasmic reticulum caused by extensive chilling stress leads to partial misfolding of proteins and its associated degradation, which eventually caused cell death of plants [61]. In soybean, flooding negatively affected the process of glycosylation of proteins related to stress glycoproteins and protein degradation while the glycolysis related proteins increased [62]. Overall accumulation of glycoproteins decreased under flooding stress. On the other hand, cold stress significantly decreased the glycosylation level of proteins especially calreticulin [14]. Under drought, the glycoproteins related to protein proteolysis increased [63]. In the present study, accumulation of glycoproteins significantly increased with silver NPs, nicotinic acid, and KNO_3_ treatment under flooding stress, suggesting the regulation of the misfolded proteins in the endoplasmic reticulum, which were damaged by the flooding stress.

## 4. Materials and Methods

### 4.1. Plant Material and Treatments

Soybean (*Glycine max* L.) cultivar Enrei was used as the plant material for the present experiment. Seeds were first surface sterilized in 2% sodium hypochlorite solution and allowed to germinate on silica sand. Seedlings were maintained at 25 °C in a growth chamber illuminated with white fluorescent light (600 μmol m^−2^·s^−1^, 16 h light period/day) and 70% relative humidity.

To study the effects of silver NPs combined with organic and inorganic chemicals under flooding stress, 15 nm silver NPs (US Research Nanomaterials, Houston, TX, USA), nicotinic acid (Sigma Aldrich, Darmstadt, Germany), and KNO_3_ (Sigma Aldrich, Darmstadt, Germany) were used. Two-day-old soybeans were treated with flooding, 5 ppm silver NPs, 8 µM nicotinic acid, 0.1 mM KNO_3_, 5 ppm silver NPs/8 µM nicotinic acid/0.1 mM KNO_3_. Concentrations of organic and inorganic chemicals were selected based on preliminary experiment data and results of previous publication [39]. Soybean were grown in seedling cases that were kept in the stainless-steel tray. After 2 days of sowing, soybean was flooded by adding water up to 4cm in a stainless-steel tray (Appendix A). After treatments, root length/weight and hypocotyl length/weight were measured at 2 days of stress. Four-day-old roots were collected for the proteomics, enzymatic, and biochemical assays. Three independent experiments were performed as biological replicates for all the experiments. Independent biological replicates were sown with one-day difference.

### 4.2. Protein Extraction

A portion (300 mg) of roots was ground with a mortar and pestle in 500 µL of lysis buffer, which contained 7 M urea, 2 M thiourea, 5% CHAPS, and 2 mM tributylphosphine. The suspension was centrifuged twice at 15,000× *g* for 10 min at 4 °C. The detergents from the supernatant were removed using the Pierce Detergent Removal Spin Column (Pierce Biotechnology, Rockford, IL, USA). The method of Bradford [64] was used to determine the protein concentration with bovine serum albumin used as the standard.

### 4.3. Protein Enrichment, Reduction, Alkylation, and Digestion

Extracted proteins (100 µg) were adjusted to a final volume of 100 µL. Methanol (400 µL) was added to each sample and mixed before addition of 100 µL of chloroform and 300 µL of water. After mixing and centrifugation at 20,000× *g* for 10 min to achieve phase separation, the upper phase was discarded and 300 µL of methanol was added to the lower phase, and then centrifuged at 20,000× *g* for 10 min. The pellet was collected as the soluble fraction [65]. Proteins were re-suspended in 50 mM ammonium bicarbonate, reduced with 50 mM dithiothreitol for 30 min at 56 °C in the dark, and alkylated with 50 mM iodoacetamide for 30 min at 37 °C in the dark. Alkylated proteins were digested with trypsin and lysyl endopeptidase (Wako, Osaka, Japan) at a 1:100 enzyme/ protein ratio for 16 h at 37 °C. Peptides were desalted with MonoSpin C18 Column (GL Sciences, Tokyo, Japan), acidified with 1% trifluoroacetic acid and analyzed by nano-liquid chromatography (LC) mass spectrometry (MS).

### 4.4. Protein Identification Using Nano-liquid Chromatography Mass Spectrometry

The LC conditions as well as the MS acquisition conditions were described in the previous study [66]. The peptides were loaded onto the LC system (EASY-nLC 1000; Thermo Fisher Scientific, San Jose, CA, USA) equipped with a trap column (Acclaim PepMap 100 C18 LC column, 3 µm, 75 µm ID × 20 mm; Thermo Fisher Scientific, San Jose, CA, USA), equilibrated with 0.1% formic acid, and eluted with a linear acetonitrile gradient (0–35%) in 0.1% formic acid at a flow rate of 300 nL min^−1^. The eluted peptides were loaded and separated on the column (NANO-HPLC capillary column C18, 3 µm, 75 µm × 150 mm; Nikkyo Technos, Tokyo, Japan) with a spray voltage of 2 kV (Ion Transfer Tube temperature: 275 °C). The peptide ions were detected using MS (Orbitrap QE plus MS; Thermo Fisher Scientific, San Jose, CA, USA) in the data-dependent acquisition mode with the installed Xcalibur software (version 4.0; Thermo Fisher Scientific, San Jose, CA, USA). Full-scan mass spectra were acquired in the MS over 375–1500 m/z with resolution of 70,000. The most intense precursor ions were selected for collision-induced fragmentation in the linear ion trap at normalized collision energy of 35%. Dynamic exclusion was employed within 15 s to prevent repetitive selection of peptides.

### 4.5. Mass Spectrometry Data Analysis

The MS/MS searches were carried out using SEQUEST HT search algorithms against the UniprotKB Glycine max (Soybean) protein database (2017-10-25) using Proteome Discoverer (PD) 2.2 (Version 2.2.0.388; Thermo Fisher Scientific, San Jose, CA, USA). Label-free quantification was also performed with PD 2.2 using precursor ions detector nodes. The processing workflow included spectrum files RC, spectrum selector, SEQUEST HT search nodes, percolator, ptmRS, and minor feature detector nodes. Oxidation of methionine was set as a variable modification and carbamidomethylation of cysteine was set as a fixed modification. Mass tolerances in MS and MS/MS were set at 10 ppm and 0.6 Da, respectively. Trypsin was specified as protease and a maximum of two missed cleavages was allowed. Target-decoy database searches used for calculation of false discovery rate (FDR) and for peptide identification FDR was set at 1%. For MS data, RAW data, peak lists and result files have been deposited in the ProteomeXchange Consortium [67] via the jPOST [68] partner repository under data-set identifiers PXD016449.

### 4.6. Differential Analysis of Proteins using Mass Spectrometry Data

The consensus workflow included MSF files, Feature Mapper, precursor ion quantifier, PSM groper, peptide validator, peptide and protein filter, protein scorer, protein marker, protein FDR validator, protein grouping, and peptide in protein. Normalization of the abundances was performed using total peptide amount mode. Significance was assessed using Abundance Ratio Adjusted P-Value. PCA was performed with PD 2.2.

### 4.7. Functional Predictions

Protein functions were categorized using MapMan bin codes [69].

### 4.8. Evaluation of Cell Death Using Evans Blue

Cell death was evaluated by Evans blue staining as described by Baker and Mock [70]. Soybean seedlings were washed twice in water and stained in 0.25% (*w*/*v*) aqueous Evans blue (Wako, Osaka, Japan) for 15 min at room temperature. The stained seedlings were washed with water and immediately photographed. For quantitative assessment of staining, the terminal 1 cm of root tip was excised and immersed in 1 mL of N,N-dimethylformamide for 24 h at 4 °C. After the incubation, the absorbance of Evans blue released into the solvent was measured at 600 nm.

### 4.9. Immuno-Blot Analysis

SDS-sample buffer consisting of 60 mM Tris-HCl (pH 6.8), 2% SDS, 10% glycerol, and 5% dithiothreitol was added to protein samples [71]. Quantified proteins (10 µg) were separated by electrophoresis on a 10% SDS-polyacrylamide gel and transferred onto a polyvinylidene difluoride membrane using a semidry transfer blotter (Nippon Eido, Tokyo, Japan). The blotted membrane was blocked for 5 min in Bullet Blocking One reagent (Nacalai Tesque, Kyoto, Japan). After blocking, the membrane was cross reacted with a 1:1000 dilution of the primary antibodies for 1 h at room temperature. As primary antibodies, the followings were used: Anti-calnexin antibody [72], anti-calreticulin antibody [73], and peroxidase-ConcanavalinA antibody (Seikagaku, Tokyo, Japan). Anti-rabbit IgG conjugated with horseradish peroxidase (Bio-Rad, Hercules, CA, USA) was used as the secondary antibody. After 1 h incubation, signals were detected using TMB Membrane Peroxidase Substrate kit (Seracare, Milford, MA, USA) following the manufacturer’s protocol. Coomassie brilliant blue (CBB) staining was used as loading control. The integrated densities of bands were calculated using Image J software (version 1.8, National Institutes of Health, Bethesda, MD, USA).

### 4.10. Statistical Analysis

The statistical significance of two groups was evaluated by the Student’s *t*-test. The statistical significance of multiple groups was evaluated by one-way ANOVA test. SPSS 20.0 (IBM, Chicago, IL, USA) statistical software was used for the evaluation of the results. A *p*-value of less than 0.05 was considered as statistically significant.

## 5. Conclusions

Nanoparticles are engineered and extensively applied on the plants to improve their growth by modulating the metabolic pathways [74]; therefore, exposing the environment to interact with these NPs [25]. Based on this information, it is essential to understand the interaction of NPs with the chemicals in the environment. The present study investigated the combined effects of silver NPs with nicotinic acid and KNO_3_ on soybean seedlings. The main findings of this study are as follows: (i) Root length/weight and hypocotyl length/weight of soybean were enhanced with silver NPs mixed with nicotinic acid and KNO_3_; (ii) protein degradation- and synthesis-related proteins increased and decreased, respectively, with silver NPs mixed with nicotinic acid and KNO_3_; (iii) the cell death was higher in the root tip under flooding stress with silver NPs mixed with nicotinic acid and KNO_3_; (iv) accumulation of calnexin/calreticulin and glycoproteins increased; and (v) growth of soybean was improved under flooding stress with silver NPs mixed with nicotinic acid and KNO_3_. In wheat, the silver NPs combined with nicotinic acid and KNO_3_ maintained the redox homeostasis through regulation of glycolysis and enhanced activities of antioxidant enzymes, which regulated energy metabolism [39]. In gladiolus, the zinc oxide NPs, which induced oxidative stress, were reduced by improving the antioxidant potential and redox homeostasis [75]. These results with previous reports suggest that the combined mixture of silver NPs, nicotinic acid, and KNO_3_ causes positive effects on soybean seedling by improving the redox homeostasis. Additionally, the regulation of protein quality control for the misfolded proteins in the endoplasmic reticulum might improve the growth of soybean under flooding stress by the combined mixture of silver NPs, nicotinic acid, and KNO_3_.

## Figures and Tables

**Figure 1 ijms-21-01300-f001:**
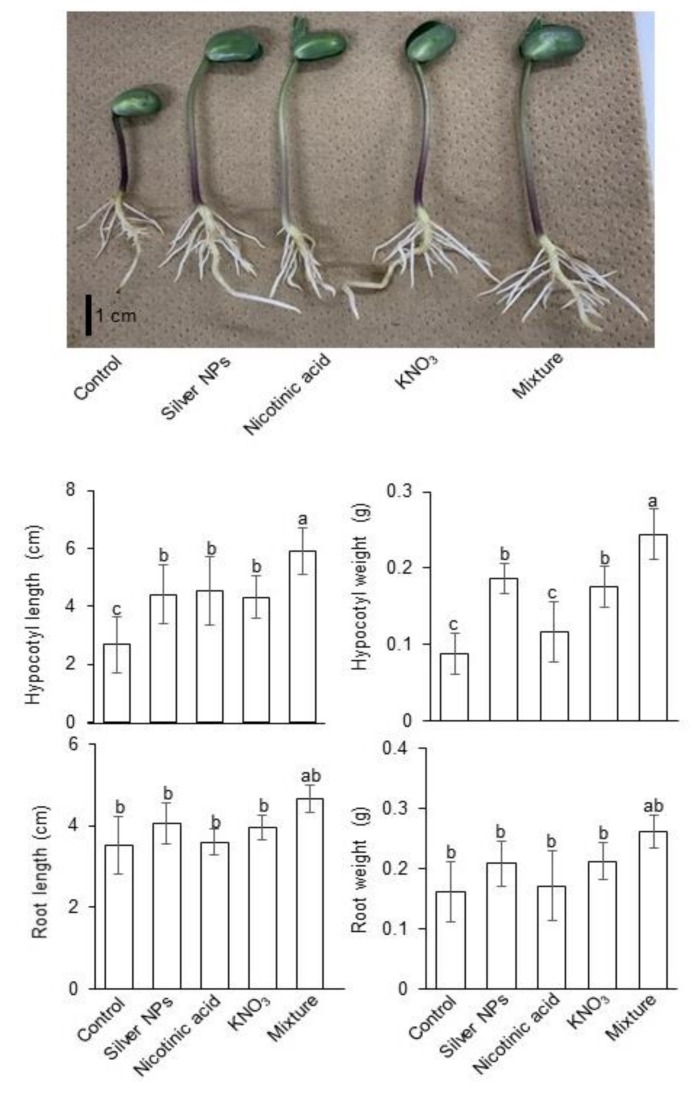
Effects of silver NPs, nicotinic acid, and KNO_3_ on the morphology of soybean seedlings. Two-day-old soybeans were treated with 5 ppm silver NPs, 8 µM nicotinic acid, 0.1 mM KNO_3_, and 5 ppm silver NPs/8 µM nicotinic acid/0.1mM KNO_3_. After treatments, root length/weight and hypocotyl length/weight were measured at two days of stress. Data are presented as the mean ± S.D. from three independent biological replicates. Mean values of each data point with different letters are significantly different according to one-way ANOVA Duncan’s Multiple Range test (*p* < 0.05). Scale bar indicates 1 cm.

**Figure 2 ijms-21-01300-f002:**
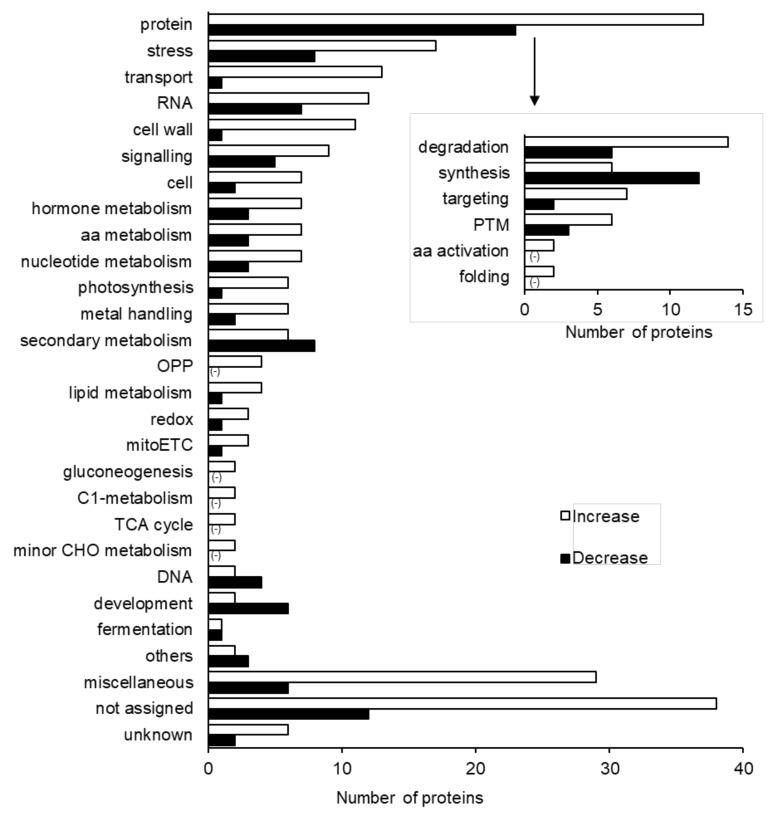
Functional categorization of proteins identified in flooding-stressed soybean treated with 5 ppm silver NPs/8 µM nicotinic acid/0.1 mM KNO_3_. Proteins extracted from root were analyzed using a gel-free/label-free proteomic technique and significantly changed proteins were identified (*p* < 0.05). The identified proteins were functionally categorized using MapMan bin codes. The x-axis indicates the number of identified proteins. Abbreviations: aa, amino acid; protein, protein synthesis/degradation/post-translational modification/targeting/folding; cell, cell division/ organization/vesicle transport; PTM, post-translational modification; RNA, RNA processing/ transcription/binding; OPP, oxidative pentose pathway; mito ETC, mitochondrial electron transport chain; C1 metabolism, carbon 1 metabolism; TCA, tricarboxylic acid cycle; CHO, carbohydrate. The negative sign shows zero proteins identified in the respective functional category.

**Figure 3 ijms-21-01300-f003:**
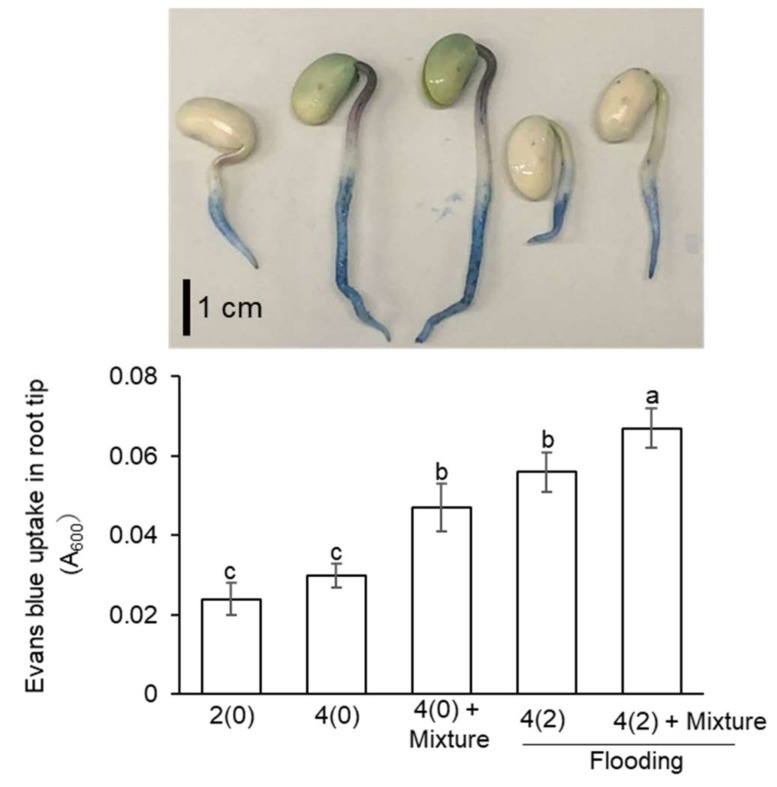
Evaluation of cell death in soybean roots treated with silver NPs/nicotinic acid/ KNO_3_ under flooding stress. Two-day-old soybeans were flooded with or without 5 ppm silver NPs/8 µM nicotinic acid /0.1 mM KNO_3_. After the treatments, soybean plants were stained with 0.25% Evans blue dye. The Evans blue dye was then extracted from the root tip and absorbance was measured at 600 nm. Data are presented as the mean ± S.D. from three independent biological replicates. Mean values of each data point with different letters are significantly different according to one-way ANOVA Duncan’s Multiple Range test (*p* < 0.05). Scale bar indicates 1 cm.

**Figure 4 ijms-21-01300-f004:**
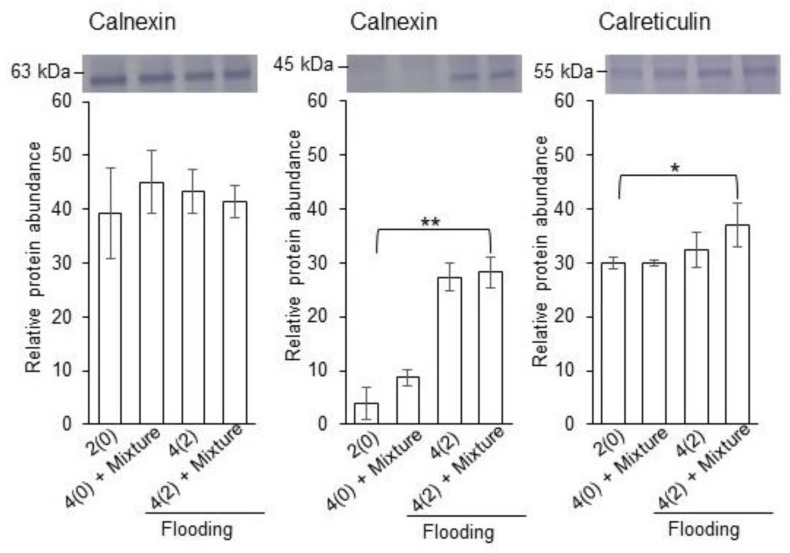
Accumulation of calnexin and calreticulin in soybean root under flooding with silver NPs/nicotinic acid/KNO_3_. Two-day-old soybeans were flooded with or without 5 ppm silver NPs/8 µM nicotinic acid/ 0.1 mM KNO_3_ for two days. Proteins (10 µg) extracted from roots were applied to gel electrophoresis and immuno-blot was performed with anti-calnexin and anti-calreticulin antibodies. The integrated densities of bands were calculated using Image J software. Data are presented as the mean ± S.D. from three independent biological replicates. Significance was calculated using Students *t*-test (* *p* < 0.05, ** *p* < 0.01).

**Figure 5 ijms-21-01300-f005:**
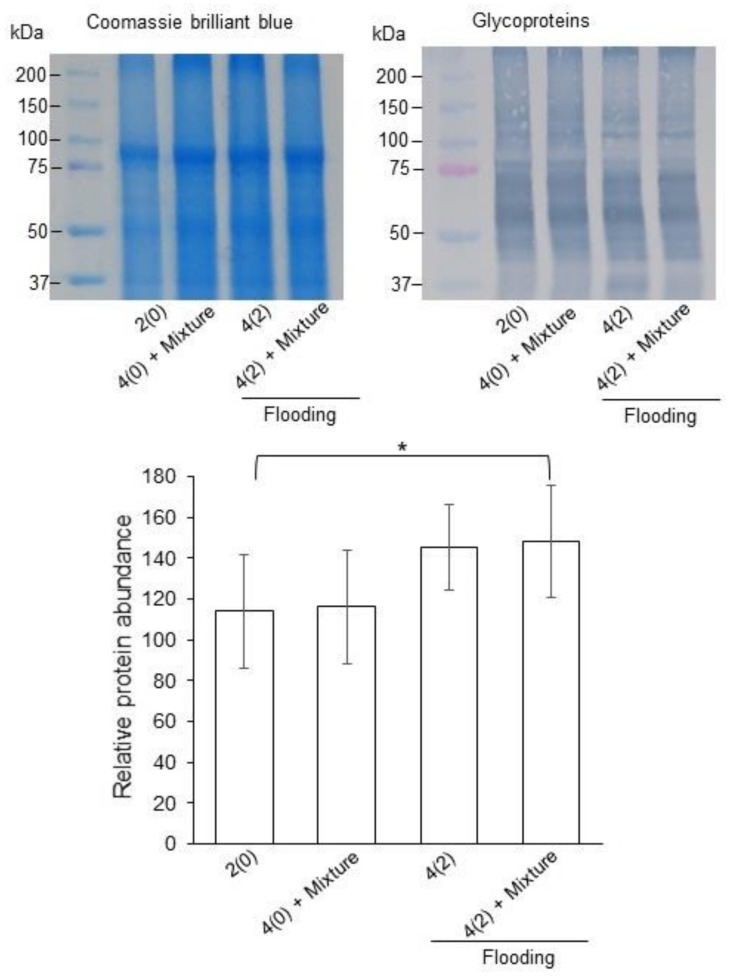
Accumulation of ConcanavalinA in soybean root under flooding stress with silver NPs/nicotinic acid/KNO_3_. Two-day-old soybeans were flooded with or without 5 ppm silver NPs/8 µM nicotinic acid /0.1 mM KNO_3_ for two days. Proteins (10 µg) extracted from roots were applied to gel electrophoresis and immuno-blotting was performed with peroxidase-ConcanavalinA antibody (left side). The integrated densities of bands were calculated using Image J software. Coomassie brilliant blue is used as a loading control (right side). Data are presented as the mean ± S.D. from three independent biological replicates. Significance was calculated using Student’s *t*-test (* *p* < 0.05).

**Figure 6 ijms-21-01300-f006:**
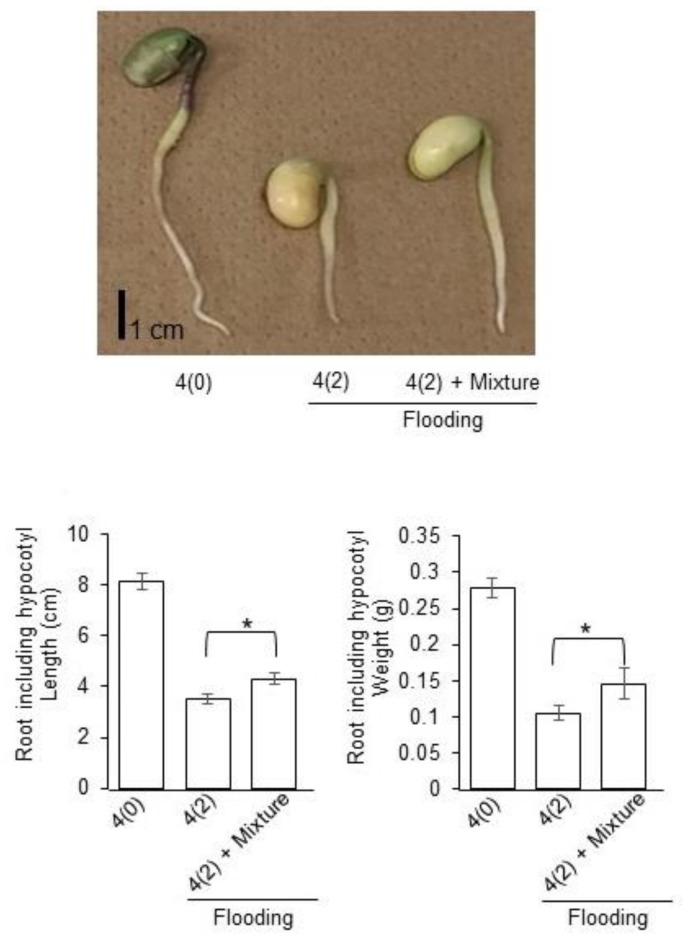
Effects of flooding stress with silver NPs/nicotinic acid/KNO_3_ on the morphology of soybean seedlings. Two-day-old soybeans were flooded with or without 5 ppm silver NPs/8 µM nicotinic acid /0.1mM KNO_3_. After treatments, length and weight of root including hypocotyl were measured at two days of stress. Data are presented as the mean ± S.D. from three independent biological replicates. Significance was calculated using Student’s t test (* *p* < 0.05). Scale bar indicates 1 cm.

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
