# Peer review of "Comparative Analysis of the Effect of Inorganic and Organic Chemicals with Silver Nanoparticles on Soybean under Flooding Stress"

_ijms, 2020, doi:10.3390/ijms21041300_

Round 1

Reviewer 1 Report

The paper was focused on comparative analysis of the effect of inorganic and organic chemicals with silver nanoparticles on soybean under flooding stress.

In my opinion, the study is interesting, however, some improvements are  recommended in order to increase its scientific soundness and contribution to the subject:

- Quality presentation of the results needs to be improved (e.g., Figure 7 - DNA bands are almost not visible in the agarose gel and there is no unit of relative DNA abundance),

- Discussion of the results as well as conclusions are superficial and should be thoroughly improved.

- English language and style should be corrected.

Author Response

Reviewer 1

The paper was focused on comparative analysis of the effect of inorganic and organic chemicals with silver nanoparticles on soybean under flooding stress.

In my opinion, the study is interesting, however, some improvements are recommended in order to increase its scientific soundness and contribution to the subject:

- Quality presentation of the results needs to be improved (e.g., Figure 7 - DNA bands are almost not visible in the agarose gel and there is no unit of relative DNA abundance),

Answer: Thank you very much for your comments. Because experiment of DNA degradation is not related to a main story of this research, Figure 7 has been removed from this article. Other figures have been added the data and re-analyzed, and then improved the figures. Furthermore, quality presentation of the result’s session has been improved. They are marked with red color.

- Discussion of the results as well as conclusions are superficial and should be thoroughly improved.

Answer: Discussion of the results and conclusion has been improved and the corrected parts have been highlighted in red color.

- English language and style should be corrected.

Answer: English language has been checked and corrected with native speaker of English.

Reviewer 2 Report

The manuscript by Hashimoto et al., investigates the combined effects of silver NPs with nicotinic acid and KNO3 on soybean seedlings. The manuscript is interesting and provides new information on the use of NPs in Soyabean. Below are my comments -

Why the authors chose NA and KNO3 to be used with silver NPs. In addition How the conc. of these chemicals were determined to be used in normal and flooded conditions. Below are my additional comments - How flooding treatment was performed? In most figures, error bars are overlapping and bars with different letters do not seem to be significantly different. The authors should check the figures and perform the stats carefully. The western blot of the Concanavalin A (fig 5) looks like a picture of the SDS-PAGE gel. Is this a blot or gel? The authors need to provide a better picture of calnexin and calreticulin immunoblots (fig 4). The bands look blurred/faint. The authors say that "detailed mechanism of soybean under the combined treatment of silver NPs, nicotinic acid, and KNO3 is still not clear". Can the authors speculate the possible mechanism of this effect and provide future directions to this study?

Author Response

Reviewer 2

The manuscript by Hashimoto et al., investigates the combined effects of silver NPs with nicotinic acid and KNO3 on soybean seedlings. The manuscript is interesting and provides new information on the use of NPs in Soyabean.

Below are my comments –

Why the authors chose NA and KNO3 to be used with silver NPs. In addition How the conc. of these chemicals were determined to be used in normal and flooded conditions.

Answer: The concentrations of NA and KNO3 were selected based on the results of previous publication, Jhanzab et al., 2019. Moreover, for the soybean, preliminary experiments were performed to select the best suitable concentration.

Below are my additional comments –

How flooding treatment was performed?

Answer: Soybeans were grown in sand in plastic seedling cases. These seedling cases were kept in the stainless steel tray. Flooding treatment was performed by adding water up to 4 cm. For the better understanding, Supplemental Figure 2 has been added to show the flooding treatment of soybean. Information has been added to the section “4.1. Plant material and treatments” and corrected parts have been highlighted in red color.

In most figures, error bars are overlapping and bars with different letters do not seem to be significantly different. The authors should check the figures and perform the stats carefully.

Answer: All figures have been re-analyzed. For Figure 1. the number of “n” has been increased and re-analyzed. Based on suggestion, Figures 4 and 5 have been improved.

The western blot of the Concanavalin A (fig 5) looks like a picture of the SDS-PAGE gel. Is this a blot or gel?

Answer: For Figure 5, as you known, because about 60% of proteins are glycoproteins, many proteins are reacted with Concanavalin A. Coomassie brilliant blue staining pattern of SDS-PAGE gel has been added in Figure 5.

The authors need to provide a better picture of calnexin and calreticulin immunoblots (fig 4). The bands look blurred/faint.

Answer: For Figure 4, pictures have been changed to other pictures.

The authors say that "detailed mechanism of soybean under the combined treatment of silver NPs, nicotinic acid, and KNO3 is still not clear". Can the authors speculate the possible mechanism of this effect and provide future directions to this study?

Answer: I am sorry that we made a mistake with writing. Sentences in the manuscript have been corrected and real meaning is as follows: “These results suggest that silver NPs synergistically enhance soybean-plant growth with nicotinic acid and KNO3 similar to the wheat. Although growth of soybean seedling significantly increased under the combined application of silver NPs, nicotinic acid, and KNO3, the detailed mechanism of soybean under the combined treatment of silver NPs, nicotinic acid, and KNO3 is still not clear. In this research, proteomic technique as well as molecular and biochemical analyses were performed to confirm the mechanism.”